# Energy Efficient Multi-Active/Multi-Passive Antenna Arrays for Portable Access Points

**DOI:** 10.3390/mi15111351

**Published:** 2024-11-01

**Authors:** Muhammad Haroon Tariq, Shuai Zhang, Christos Masouros, Constantinos B. Papadias

**Affiliations:** 1Smart Wireless Future Technologies (SWIFT) Lab, Under the Research Technology and Innovation Network (RTIN), The American College of Greece (ACG), Ag. Paraskevi, 153 42 Athens, Greece; cpapadias@acg.edu; 2The Department of Electronics Engineering in Antennas, Propagation and mmWave Systems (APMS) Section, Aalborg University, 9220 Aalborg, Denmark; sz@es.aau.dk; 3Information and Communication Engineering Group, University College London (UCL), London WC1E 6BT, UK; c.masouros@ucl.ac.uk

**Keywords:** beamforming, multi-active/multi-passive (MAMP) antenna arrays, multi-input multi-output (MIMO), hybrid antenna arrays, portable access points (PAPs), radiation conditions, radiation efficiency

## Abstract

This article is about better wireless network connectivity. The main goal is to provide wireless service to several use cases and scenarios that may not be adequately covered today. Some of the considered scenarios are home connectivity, street-based infrastructure, emergency situations, disaster areas, special event areas, and remote areas that suffer from problematic/inadequate network and possibly power infrastructure. A target system that we consider for such scenarios is that of an energy-efficient self-backhauled base station (also called a “portable access point—PAP”) that is mounted on a drone to aid/expand the land-based network. For the wireless backhaul link of the PAP, as well as for the fronthaul of the street-mounted base station, we consider newly built multi-active/multi-passive parasitic antenna arrays (MAMPs). These antenna systems lead to increased range/signal strength with low hardware complexity and power needs. This is due to their reduced number of radio frequency chains, which decreases the cost and weight of the base station system. MAMPs can show a performance close to traditional multiple input/multiple output (MIMO) systems that use as many antenna elements as RF chains and to phased arrays. They can produce a directional beam in any desired direction with higher gain and narrow beamwidth by just tuning the load values of the parasitic elements. The MAMP is designed based on radiation conditions which were produced during the research to ensure that the radiation properties of the array were good.

## 1. Introduction

The use of portable access points (PAP’s) or unmanned aerial vehicles (UAV’s) serving as data relays/base stations holds significant importance for delivering on-demand connectivity as well as providing public safety services or aiding in recovery after communication infrastructure failures caused by natural disasters. PAPs could be utilized to address this issue by mounting small cell base stations on them. The deployment of such base stations can face some restrictions that need to be considered, such as the availability of reliable wireless backhaul links to transfer all the traffic to the core network from the targeted area, see scenario setup in Figure 1. These small cells are self-backhauled, and they can be mounted on a drone. In the case of a disaster, the whole network could be out of service, resulting in an outage of communication links. The demand for high speed and reliable communication has brought new solutions in terms of 5G and beyond, where most of the focus is on using small-cell deployments due to their shorter wavelengths (mmWaves) for self-backhauled base stations. In order to satisfy the needs of the proposed network, a base station is considered that benefits from highly directional antennas with narrow beamwidth and adaptive precoding schemes. Conventional methods of achieving high-gain narrow beams involve a huge number of antenna elements, thus increasing the number of RF chains. Mounting this antenna system on a drone will increase the weight and battery usage of the drone.

A conforming antenna system is presented in [1], where 1800 coverage is achieved by using three antennas on one arm of the UAV. A comprehensive survey of patch antennas for UAVs is given to show the potential of patch antennas for drones. Dual band antennas have been of great interest for the scientific community in that they can be used for backhauling and access links simultaneously [2,3]. Patch antennas and the array do not seem like suitable solutions for high directional beams due to their increased size and the weight of the antenna system. A high-gain reconfigurable antenna system [4] is designed for the connection between a drone and ground connectivity. Switching modes can provide different beam directions and polarizations but at the cost of large size and weight. The number of RF chains are also huge due to the overall coverage needed for ground connectivity. Comparison of a circularly polarized patch antenna with a dipole is presented to show the enhanced performance of patch antennas using a protective structure for a drone [5]. Multiantenna interference technology is enabled to show the interference at any position in a 2D plane [6].

To cope up with the challenges of wireless backhaul, microwave and mmWave massive MIMO communication is considered for hybrid transceivers. To reduce the complexity of RF chains being used in beamforming, parasitic antenna arrays are considered that may include a combination of active antenna elements and parasitic elements tuned by loads. MIMO transmission has attracted a plethora of researchers and scientists in recent years due to its benefits for spatial multiplexing and large antenna and diversity gain. These benefits increase as the number of antennas increases; however, this comes at the expense of the size, cost, and complexity of the required radiofrequency (RF) chains. The size and cost of multiple antenna systems have been reduced by utilizing a single RF chain as of several years ago, when the idea was first introduced by Harrington [7]. The research community has since significantly contributed to the concept of Electronically Steerable Parasitic Array Radiators (ESPAR). If parasitic elements are placed in the close vicinity of active elements, then the induced currents may affect the beam pattern generated by the array [8,9].

Many different techniques and approaches have been proposed in the past to achieve beamforming using ESPARs, such as: a Hamiltonian method [10], an algorithm based on sequential perturbation [11], a cross-correlation coefficient maximization technique [12], and simultaneous perturbation stochastic approximation (SPSA) [13]. A technique for spatial multiplexing of QPSK symbols has been introduced by Bains and Kalis, which is known as the beam space model [14,15,16,17]. Some applications have also been proposed using ESPAR, for example, the Reactance Domain MUltiple SIgnal Classification (RD-MUSIC) algorithm [18], a system for the implementation of direction of arrival (DOA) estimation based on a rotational invariant technique (ESPRIT) [19]. In the early approaches of ESPARs, it was ensured that the array radiates based on the estimation of loads while neglecting the radiation conditions being incorporated in the algorithms.

Design guidelines and radiation constraints were introduced for the first time in [20] to ensure that the single-active/multi-passive (SAMP) array will radiate. Based on these radiation conditions, researchers proposed solutions for different communication schemes e.g., MIMO and beamforming using single RF chain [21,22,23,24,25,26,27,28]. Although authors developed an algorithm for 3D beamforming using MAMPs [29] previously developed in [24] but radiation conditions were not met for the MAMP arrays. Mutual coupling seriously affects the multi antenna performance if elements are placed in close vicinity. To address this issue radiation conditions for hybrid multi-active/multi-passive (MAMP) antenna arrays were developed [25]. The conditions determine whether the MAMP array will radiate as whole or if some of the elements will radiate. Examples show that even if an array produces a radiation comparable to the desired beam, there might be a possibility that the array as a whole will radiate or there might be a possibility that any of the antennas will not radiate. This points to a need for the quantification of the radiation efficiency of MAMP arrays. In this work, a mathematical expression for the quantification of the total radiation efficiency of MAMP arrays is derived in Section-I (b). An algorithm is developed in Section-I (b) for the computation of the parameters of MAMP arrays that produce a desired radiation pattern subject to radiation efficiency conditions. The CST simulation results and the theoretical analysis from MATLAB 2020a are presented in Section-II (b). A 2-active 12-passive MAMP array prototype is developed and a comparison of simulated and measured results is shown in Section-II (c).

### 1.1. System Model

The system consists of multiple RF chains being fed by a vector of voltages $v_{M}$, which generates currents i at each element of the array. Each active element is related to respective parasitic elements by the mutual coupling induced by the currents of the neighboring elements, thus producing a SAMP. Multiple SAMPs are placed at a distance of λ/2, resulting in a MAMP array as shown in Figure 2, which is based on the mutual coupling among the elements represented as $Z_{MM}$. Parasitic elements are placed at a distance of *d* which is a function of λ, depending on the combination of loads that produce a radiation pattern close to the ideal or desired beam (which is a beam produced by an equivalent number of elements in a ULA).

The elements are placed in close vicinity to the active elements to achieve a strong mutual coupling through strong induced currents. The generated currents for the proposed system model are given as follows:(1)$$i={\left[Z_{MM}+Z_{L}\right]}^{-1}v_{M}$$
where i is the vector representing currents at all elements of the MAMP array. $Z_{L}$ is a matrix representing the source resistance at all the active elements and the load values of parasitic elements. $Z_{MM}$ is the mutual coupling matrix and represents the coupling among the elements of the whole array. It shows the self-impedance and the mutual impedance of the neighboring elements and can be seen in Equation (2). $v_{M}$ is the input voltage at the feeding active elements of the arrays and at the parasitic elements of the array, which is zero. Equation (1) can be expanded for the MAMPs as given in [25] as follows:(2)$$\left[\begin{array}{ccc} Z_{11}+x_{1} & Z_{12} & \cdots \\ Z_{21} & Z_{22}+R_{1} & \cdots \\ Z_{31} & Z_{32} & \cdots \\ \vdots & \vdots & \ddots \\ \vdots & \vdots & \ddots \\ \vdots & \vdots & \cdots \\ Z_{M1} & Z_{M2} & \cdots \end{array}\begin{array}{c} Z_{1M}\\ Z_{2M}\\ Z_{3M}\\ \vdots \\ \vdots \\ \vdots \\ Z_{MM}+x_{M} \end{array}\right]\left[\begin{array}{c} i_{1}\\ i_{2}\\ i_{3}\\ \vdots \\ \vdots \\ \vdots \\ i_{M} \end{array}\right]=\left[\begin{array}{c} 0\\ v_{1}\\ 0\\ \vdots \\ 0\\ v_{N}\\ 0 \end{array}\right]$$

From (2), the variable $x_{M}$ is the *m*-th load value of the respective parasitic element of the array, where $i_{m}\;$ is the current at the *m*-th element and $v_{j}$ is the voltage at the *j*-th active element in the MAMP array, and *j* ϵ {1, 2, 3, …,N}. Since each set of single-active/multi-passive (SAMP) array is placed at half a wavelength’s distance from every other SAMP array, the active elements are half a wavelength from each other, thus reducing unwanted coupling effects. Conversely, the parasitic elements are placed in close proximity to receive the most coupling. The performance of the multiport antenna array can be analyzed by using the input impedance of the ports. The input impedance connected to the RF chains incurring the effect of mutual coupling from neighboring elements can be obtained from (2) as follows:(3)$$Z_{in}^{j}=\sum_{m=1}^{M}Z_{jm}\frac{i_{m}}{i_{j}},\;j\epsilon \left\{1,\;2,\;3,\;\ldots ,\;N\right\}$$
where $Z_{in}^{j}$ represents the input impedance at the *j*-th port, $Z_{jm}$ is the mutual coupling between the *j*-th active element and the *m*-th element in the MAMP array, and $i_{j}$ represents the current at the *j*-th active element in the MAMP array.

### 1.2. Radiation Conditions

The expression in (3) has been used as a constraint in [25] to ensure the radiation of the MAMP array. The mutual coupling among the elements can drive the real part of the input impedance to a negative value i.e., $Z_{in}^{j}<0,\;$ which means that the power is not radiated by the array and is dissipated instead. The radiation condition for a MAMP array is given by [25]:(4)$$-Re\left\{\sum_{m=1,\;m\neq j}^{M}Z_{jm}\frac{i_{m}}{i_{j}}\right\}<Re\left\{Z_{jj}\right\}$$

So, for the array to radiate, the input impedance at all the feeding ports must be greater than zero, or it must be a positive number that remains greater than the self-impedance $Z_{jj}$ of the respective active element. The developed algorithm makes sure that the input impedance at all the feeding ports is greater than zero, which ensures that the array radiates as a whole.

These radiation conditions do not satisfy the desired array performance, because in [25], constraints guarantee array radiation when either of the active port is radiating or all the elements are radiating based on the comparison of radiation patterns, regardless of the overall array efficiency. This means that the desired radiation pattern can be achieved, but the radiation efficiency of the array might be affected due to power dissipation at any of the feeding ports. Radiation efficiency is one of the most significant parameters of a multi-port antenna array that indicates reliable overall array performance. The radiation efficiency of an antenna can be defined as the ratio of total radiated power to the total input power. So, from [30,31] we know that the radiation efficiency of the antenna array is as follows:(5)$$\eta =\frac{P_{rad}}{P_{in}}$$
(6)$$\Gamma_{T}=\sqrt{1-\frac{P_{rad}}{P_{in}}}$$

A total active reflection coefficient (TARC) is introduced in [31] and is given by $\Gamma_{T}$, which, in turn, can be written in terms of radiation efficiency. The TARC is considered as the most valuable parameter to analyze the performance of a multiport array because it considers the mutual coupling among the elements and takes into account the weights of the voltages, which are in the form of different amplitudes and phases. From (5) and (6), we can obtain the following:(7)$$\eta =1-{\Gamma_{T}}^{2}$$
where $\Gamma_{T}$ for two active ports can be defined as follows:(8)$$\Gamma_{T}=\sqrt{\frac{{\left|V_{1}S_{11}+V_{2}S_{12}\right|}^{2}+{\left|V_{1}S_{21}+V_{2}S_{22}\right|}^{2}}{{\left|V_{1}\right|}^{2}+{\left|V_{2}\right|}^{2}}}$$

Using (8) in (7), the total efficiency of the array can be calculated, but given that TARC uses S-parameters, the radiation conditions cannot be integrated without a conversion from S- to Z-parameters. The S-parameters can be converted to Z-parameters using some conversion expressions, the formulae for which are given for a two port network by Pozar in [9] as follows: (9)$$S_{11}=\frac{\left(Z_{11}-Z_{0}\right)\left(Z_{22}+Z_{0}\right)-Z_{12}Z_{21}}{\left(Z_{11}+Z_{22}\right)\left(Z_{22}+Z_{0}\right)-Z_{12}Z_{21}}$$
(10)$$S_{12}=\frac{2Z_{12}Z_{0}}{\left(Z_{11}+Z_{22}\right)\left(Z_{22}+Z_{0}\right)-Z_{12}Z_{21}}$$
(11)$$S_{21}=\frac{2Z_{21}Z_{0}}{\left(Z_{11}+Z_{22}\right)\left(Z_{22}+Z_{0}\right)-Z_{12}Z_{21}}$$
(12)$$S_{22}=\frac{\left(Z_{11}+Z_{0}\right)\left(Z_{22}-Z_{0}\right)-Z_{12}Z_{21}}{\left(Z_{11}+Z_{22}\right)\left(Z_{22}+Z_{0}\right)-Z_{12}Z_{21}}$$

The generalized form of the above conversions for multiple antenna elements is obtained using (2) as follows:(13)$$S_{ii}=\frac{\left(\left(Z_{NN}+x_{k}\right)-Z_{0}\right)\left(Z_{MM}+Z_{0}\right)-Z_{NM}Z_{MN}}{\left(\left(Z_{NN}+x_{k}\right)+Z_{0}\right)\left(Z_{22}+Z_{0}\right)-Z_{NM}Z_{MN}}$$
(14)$$S_{ij}=\frac{2Z_{NM}Z_{0}}{\left(\left(Z_{NN}+x_{k}\right)+Z_{0}\right)\left(Z_{MM}+Z_{0}\right)-Z_{NM}Z_{MN}}$$
(15)$$S_{ji}=\frac{2Z_{MN}Z_{0}}{\left(\left(Z_{NN}+x_{k}\right)+Z_{0}\right)\left(Z_{22}+Z_{0}\right)-Z_{NM}Z_{MN}}$$
(16)$$S_{jj}=\frac{\left(\left(Z_{NN}+x_{k}\right)+Z_{0}\right)\left(Z_{MM}-Z_{0}\right)-Z_{NM}Z_{MN}}{\left(\left(Z_{NN}+x_{k}\right)+Z_{0}\right)\left(Z_{22}+Z_{0}\right)-Z_{NM}Z_{MN}}$$

The extended form of the expressions is given in (13) to (16) to compute S-parameters among the elements of the MAMP array, where $i\;\epsilon \;\left\{1,\;2,\;3,\;\ldots ,\;N\right\}$, $\;j\;\epsilon \;\left\{1,\;2,\;3,\;\ldots ,\;M\right\},\;and\;i\neq j.$ $x_{k}\;$ is the k-th load value of the respective parasitic element. For an ideal array, from (8), the radiation efficiency of the array will be one, but in the case of a MAMP array, there will be coupling among the neighboring elements, so it will be a number between zero and one. The constraint shows the η of the array as one if the total active reflection coefficient is zero.
$$\begin{array}{c} \mathrm{So},\;\mathrm{the}\;\mathrm{new}\;\mathrm{constraint}\;\mathrm{is}\;0\;<\;\eta \;<\;1\\ \mathrm{If}\;\Gamma_{T}=0,\;\mathrm{then}\;\eta =1 \end{array}$$

For any desired η for a MAMP array with *N* number of elements, the efficiency of the MAMP array is given by the following formula:(17)$$\eta =1-\sqrt{\frac{\;\sum_{i=1}^{N}\;\left({\left|\sum_{j=1}^{N}v_{j}S_{ij}\right|}^{2}\right)}{\sum_{i=1}^{N}{\left|v_{i}\right|}^{2}\;}}$$

The alternating optimization approach is used for the stochastic beamforming algorithm [24] by introducing the new radiation conditions and a newly developed cost function. Newly developed Algorithm 1 guarantees that the array is radiating as a whole, and it computes the load values for a MAMP array which has optimized radiation efficiency. The cost function that we attempt to optimize can be viewed as a function of both the loads and the voltage vector. Thus, for a selected antenna array structure that determines the set S of AEs and the impedance matrix Z, the new cost function must be minimized and is given by the following:(18)$$\begin{array}{c} L\left(x_{Sc},w\right)≔\omega \left[1-\left(\;\frac{\left|b^{H}a\left(x_{Sc}\;,w\right)\right|}{{\left\|b\right\|}_{2}{\left\|a\left(x_{Sc}\;,w\right)\right\|}_{2}}\right)\right]+µ\left[1-\left(\;\eta \left(x_{Sc}\;,w\right)\right)\right]\\ \mathrm{Where}\;\omega +µ=1, \end{array}$$

ω is the weight for the radiation pattern, and µ is the weight for the radiation efficiency,
$$\begin{array}{c} w=\left[\begin{array}{c} real\left(v_{S}\right)\\ imag\left(v_{S}\right) \end{array}\right],and\\ a\left(x_{Sc}\;,w\right)=S^{T}{\left(Z+diag\left(I_{S}x_{S}+jI_{Sc}x_{Sc}\right)\right)}^{-1}v\left(w\right),\\ v\left(w\right)=I_{S}{(w}_{1:N_{a}}+jw_{\left(N_{a}+1\right):2N_{a}}),\\ S=\left[S\left(\phi_{1}\right),\ldots ,\;S\left(\phi_{N}\right)\right] \end{array}$$

**Algorithm 1** Alternating Optimization Stochastic Beamforming Algorithm1: **function** AO-SBA ((b, ${\left(\beta_{m}\right)}_{m=1}^{M}$, *S*, **Z**, **S**, *τ*, *N_m_*, *T_er_*, *eps*,  ω, µ)
2: $\;m\;\leftarrow 0,x_{Sc}={\left[0,\ldots ,0\right]}^{T\;},w\leftarrow {\left[1^{T},0^{T}\right]}^{T}$
3: **while** m<M **do**
4:   m ← m + 1, n ← 0, k ← 0
5: ${\;\;err}_{x}$ ← 1/eps, ${err}_{w}$ ← 1/eps
6:


**while** $n<N_{m}$ **and** ${err}_{x}\geq T_{er}$ **do**
7:        n ← n + 1, $x_{old}$ ← $x_{Sc}$
8:       Create: $\delta_{x}$ ∼ B(1, 1/2) with values ±1
9:       $x_{Sc}^{+}=x_{Sc}+\beta_{m}\delta_{x}$, $x_{Sc}^{-}=x_{Sc}-\beta_{m}\delta_{x}$
10:     $L_{x}^{+}=L\left({(x}_{Sc}^{+}\;,w)\right),\;L_{x}^{-}=L\left({(x}_{Sc}^{-}\;,\;w)\right)$
11:     $\xi_{x}=\frac{1}{2\beta_{m}}\;\left(L_{x}^{+}-L_{x}^{-}\right)\;1⊘\delta_{x}$
12:     $x_{Sc}=x_{Sc}-\tau \xi_{x},\;{err}_{x}=\frac{{\left\|x_{Sc}-x_{old}\right\|}_{2}}{{\left\|x_{old}\right\|}_{2}+eps}$
13: **end while**
14: **while** k<Nm **and** errv≥Ter **do**
15:       k ← k + 1, wold ← w16:       Create: δw ∼ B(1, 1/2) with values ±117:       w+=w+βmδw, w−=w−βmδw18:       Lv+=LxSc ,w+, Lx−=LxSc ,w−19:       ξw=12βm Lv+−Lv−1⊘δw20:       w=w−τξw, errw=w−wold2wold2+eps21: **end while**22: **while**  real Zinj>0, **do** Given by (3.3) and (3.4)
23:   p ← p + 1, jϵ1, 2, 3, …,N24: Go to **function** AO-SBA25:      **while** 0 < η < 126:                p ← p + 127:                ω+µ=1,28:                η=1−ΓT2, ΓT=1−PradPin29:      **end while**30: **end while**31:   Erm← LxSc ,w  Given by (3.18) and (3.19)32: **end while**33: **end function****Ensure:** Er1:M, xSc, w,η


S is the steering matrix of the MAMP antenna and b is the radiation pattern of the desired beam pattern. The optimization in (18) is non-linear with respect to both variables $x_{Sc}$ and w. Therefore, we attempt to solve the optimization task with an alternating optimization stochastic approach. The AO-SBA algorithm is following an iterative approach to find the optimized loads and the weights with reduced error, whereas the degree of cost function smoothing is achieved by using the smoothing sequence ${\left(\beta_{m}\right)}_{m=1}^{M}$. The process repeats M times for different smoothing steps until the local minimal probability is reduced. A random vector δ is created using binomial distribution, then the cost function samples are computed. This leads to the approximation of the gradient until its convergence. This estimates the approximation and returns the estimated solution; otherwise, the iteration counter is increased by one. The global minimization corresponds to a radiation pattern identical to the desired pattern, but the convergence time will be expanded, since, in this case, the smoothing sequence should be larger, increasing the number of required cycles until the termination of the overall process [26]. The estimated solution provides the computed loads, which give the corresponding input impedance at each port. The next cost function, the real part of the input impedance which must be greater than zero, is achieved by many iterations, and the last step of the algorithm is to find the efficiency of the array by finding the TARC.

## 2. MAMP Design and Results

### 2.1. Computation of Loads

A Multi-Active/Multi-Passive parasitic antenna array is considered with 2 active and 12 parasitic elements, as shown in Figure 3. The loads values are obtained using SBA with new radiation conditions such that the whole array radiates. The desired radiation efficiency of the array is calculated by using (17). There is a possibility that the SBA generates negative load values (negative real part of the loads means negative resistive part) which is difficult to implement in real time. So, the algorithm is modified in such a way that it ensures the following four steps:

the algorithm computes the radiation efficiency of the array by using the given parameters,it compares the radiation patterns of the proposed MAMP array with that of the ideal radiation pattern produced by the ULA,it generates the optimized passive load values,it performs checks to satisfy all the radiation conditions which ensure that the whole array radiates, and the input impedance is greater than zero at all feeding ports.

The resonant frequency is 2.4 GHz, and the elements are considered as point sources for the calculation of loads. The mutual coupling and the self-impedance of the elements are calculated using Balanis’s equations, which are based on point sources. The active elements are placed half a wavelength apart, whereas the parasitic elements are placed at 0.22*λ from each other. The elements must be placed close enough so that the effect of mutual coupling constructs in such a way that it produces higher currents at the neighboring elements. Many iterations were carried out before finalizing the inter-element distance. An optimized value is obtained after multiple calculations that satisfies all the conditions and constraints.

The optimal value may not produce the realistic values of the loads, and it may not have enough efficiency, so the algorithm has to go through more iterations. The tradeoff between efficiency and the ideal radiation pattern exists and it can be obtained by first obtaining the error in terms of the comparison between the radiation pattern of the proposed MAMP array and the radiation pattern of the ideal ULA. This can be controlled by using the weights as defined in the cost function, see (18). This pattern is achieved after many iterations of the algorithm based on inputting specific sequence and parameter variation in the algorithm to obtain the optimized loads and weights for the required radiation efficiency and radiation pattern. the optimal values are M = 40, τ = 100, MAXITER = 10,000, $T_{er}$ = 10^−6^, eps = 10^−10^, ω = 0.65, and μ = 0.35.

The sequence and variation parameters might have to change to obtain optimized results for different angles to achieve beam steering. As a tradeoff exists between radiation efficiency and the radiation pattern, there is also a tradeoff between the difference in the values of the above parameters. It is very difficult to achieve the optimized and desired results with the same values of the parameters. So, in order to obtain beam steering, the distance between the elements can be changed, but it is a big challenge to find the beam steering angles with a fixed distance among the elements. The sequence parameters play a crucial role in computing the load values for the desired radiation efficiency and the desired radiation pattern. After many iterations and computations, the proposed distance is obtained which ensures beam steering at different angles with a fixed inter-element distance. Another challenge is to find out the optimized passive load values and the realistic load values. The sequence parameters control the limit of digits in the load values, which makes it either possible or not to implement the parasitic antenna array. A prior knowledge of the values for passive components existing in the market is necessary to obtain the easily available load values.

### 2.2. Simulations Using CST Microwave Studio

The MAMP array setup as shown in Figure 3 is emulated in CST microwave studio to achieve the emulated results of the proposed antenna array. A geometry of the array is shown in Figure 4, where finite-length, half-wave dipoles are considered to develop the proposed MAMP array in CST, as it is not possible to simulate a point source in simulation tools. The thickness of the dipoles is decided after a survey of which copper tubes are commercially available. The frequency of the operation is 2.4 GHz. FR-4 is used as a substrate to support the copper cylinders. The active elements are being fed by two microstrip feed lines and are excited by using SMA connectors. Both of the active monopoles are fed by using SMA connectors through microstrip lines and are grounded partially, whereas all the monopoles are embedded on the top layer with copper pads and are grounded through via holes. The monopoles on the top are connected to passive components and are embedded on the pads at the top layer, as shown in Figure 5a. These passive components are then grounded through via holes and connected to the grounded monopoles as shown in Figure 5b. CST simulations are performed using the time domain and the integral solvers, whereas the CST license is provided by Aalborg University Denmark.

A comparison of the radiation patterns is shown in Figure 6 (2D polar plots) and Figure 7 (3D radiation patterns). The proposed MAMP is compared with a 5 active element uniform linear array (ULA). The curve in red represents the ideal radiation pattern of a 5-dipole ULA. The magenta represents the ideal beam of a ULA, but with the same number of elements as the number of active elements in MAMP. The radiation pattern computed by SBA for the MAMP is presented in blue, whereas the simulated radiation pattern of the MAMP is represented in black. This Figure can be explained in three steps:

First, the comparison between the MAMP and the ULAs shows that the desired radiation pattern or beam can be obtained by reducing the number of active elements, and thus the number of RF chains. Even with a reduced number of active elements, we can still obtain a beam which is similar to that of conventional ULAs.Then, the comparison is between the simulated and the calculated beams by CST and the algorithm. The side lobes in simulated results are bigger than the calculated beams. This is most likely due to the non-ideal finite length designs in the simulation tool. The calculated beams are based on ideal point sources, whereas ideal simulations are not possible in the simulation tool. Sometimes, residues add up to give higher values at the end of each iteration.

### 2.3. MAMP Antenna Array Prototype

The MAMP antenna array prototype was developed at University College London (UCL). Figure 8 shows the developed antenna, where copper dipoles are embedded on a PCB where the substrate is FR-4. The copper monopoles are soldered on each side of the PCB board and the parasitic elements are soldered and grounded through SMD passive components and using via holes. Active dipoles are fed through microstrip lines and SMA connectors. The developed prototype is tested using a network analyzer to see the resonance of the antenna. Figure 9 shows the setup for the measurements of the S-parameters. The antenna ports are connected to the power splitter/combiner and the input of the VNA is connected to the splitter’s input port. The MAMP antenna was also tested in an anechoic chamber in a facility at Intracom Defense (IDE), Greece. The radiation pattern of the prototype was measured in the chamber. The measurements setup and the developed antenna are shown in Figure 8, Figure 9 and Figure 10.

The simulated and measured S-parameters show a good match, as shown in Figure 11. There is a slight shift in the resonant frequency of the prototyped antenna, as shown in the measured S-parameters, but it is still well-matched for the desired frequency, i.e., 2.4 GHz. A slight discrepancy is due to the use of a wideband power splitter while performing measurements. Figure 12 shows both the simulated and measured radiation patterns of the antenna array. Both the simulated and measured radiation patterns are similar in shape, except for the fact that the measured results show a smaller beamwidth as compared to the simulated pattern. The peak gain of the simulated antenna is 7.75 dBi, whereas the peak gain of the measured antenna is 6.9 dBi, as shown in Figure 13. The difference in the peak gains could be due to cable losses and the connectors. It can also be due to losses at passive components. The simulated radiation efficiency of the MAMP is 98%. Both the simulated and measured results depict excellent performance of the MAMP array at the given frequency. The beamforming is also shown with the development of hardware design. The peak gain of the proposed antenna can be further enhanced by placing a reflector to reduce the back lobe. It can also be enhanced by adding a greater number of parasitic elements and then furthering the active element as per the requirements of the application.

## 3. Conclusions

An energy-efficient antenna array was built to ensure reliable backhaul connection from air-to-ground and air-to-air connectivity. A detailed comparison of proposed MAMP array with the previous study is presented in Table 1. The antenna array is cost- and weight-effective due to its smaller number of RF chains compared to any conventional directional antenna array. The antenna array consists of few active and parasitic antenna elements benefiting from mutual coupling. A mathematical expression for the radiation efficiency of MAMP arrays was developed. A new cost function was introduced by integrating radiation conditions and radiation efficiency. An algorithm was developed to compute the load values. A radiation pattern similar to all-active ULA is achieved with MAMP array with 3 times fewer RF chains. A 2A-12P MAMP array prototype was developed and tested in an anechoic chamber. The measured radiation pattern matches well with the simulated results. The measured peak gain of the developed MAMP antenna array is 7.5 dB (with 2 RF chains). The developed MAMP’s radiation efficiency exceeds 90% at 2.4 GHz. The new MAMP antenna array theoretical framework allows for the design of hybrid arrays with performance similar to conventional arrays with significantly reduced hardware, size, and cost. These features make MAMP arrays particularly appealing for portable devices that are constrained in terms of size and energy, such as UAV-borne portable access points. The development of this MAMP antenna array for MIMO systems could be a possible extension of this work with more focus on finding the currents at each element of the MAMP rather than finding the loads at parasitic elements. Current work considered ideal point sources for the calculation of coupling among the elements, meaning a new focus can be the distribution of currents at each element and finding the efficiency of the MAMP antenna array system.

## Figures and Tables

**Figure 1 micromachines-15-01351-f001:**
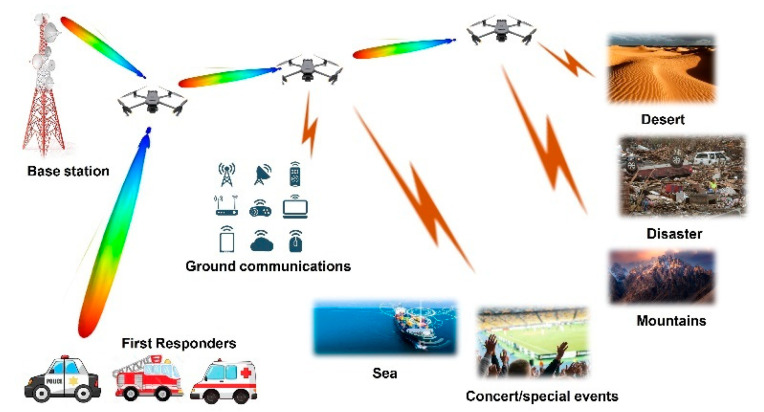
Backhauling of drones and connectivity for emergency situations.

**Figure 2 micromachines-15-01351-f002:**
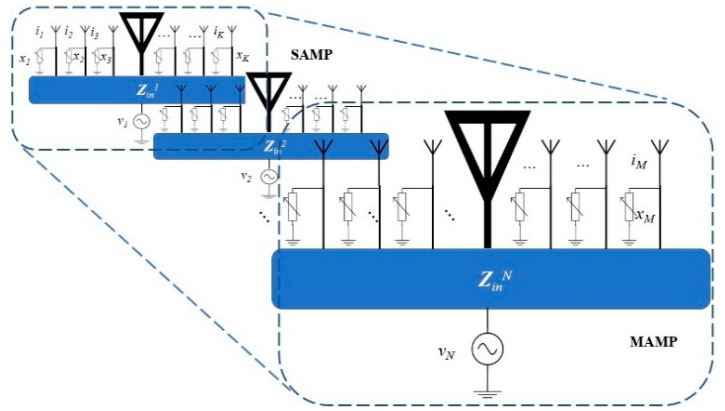
Multi-Active/Multi-Passive parasitic antenna array geometry.

**Figure 3 micromachines-15-01351-f003:**
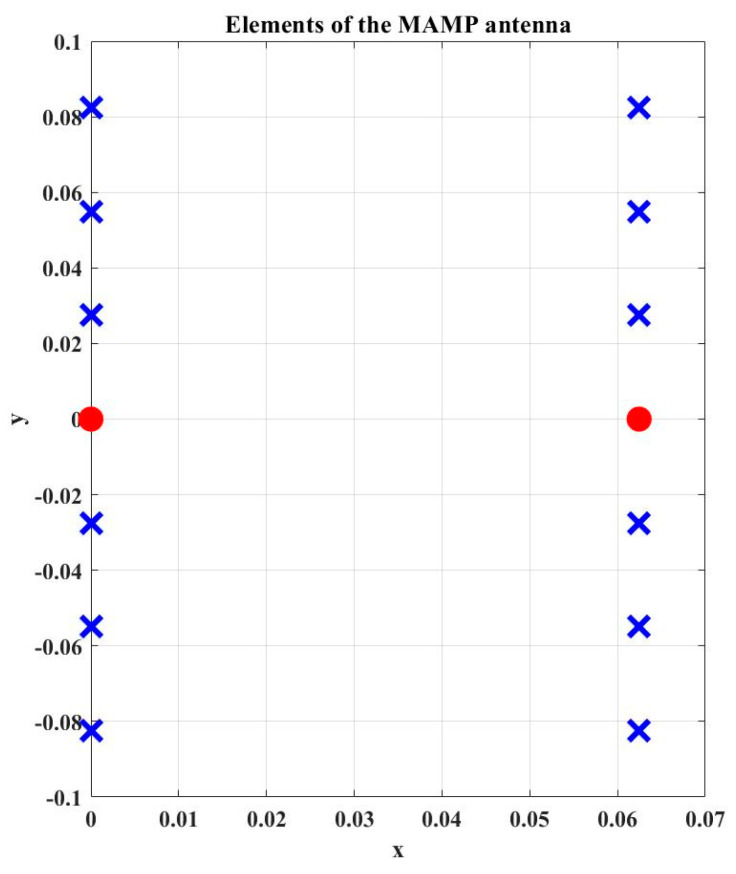
MAMP geometry in MATLAB; blue colored crosses represent the parasitic elements and the red dots represent the active elements.

**Figure 4 micromachines-15-01351-f004:**
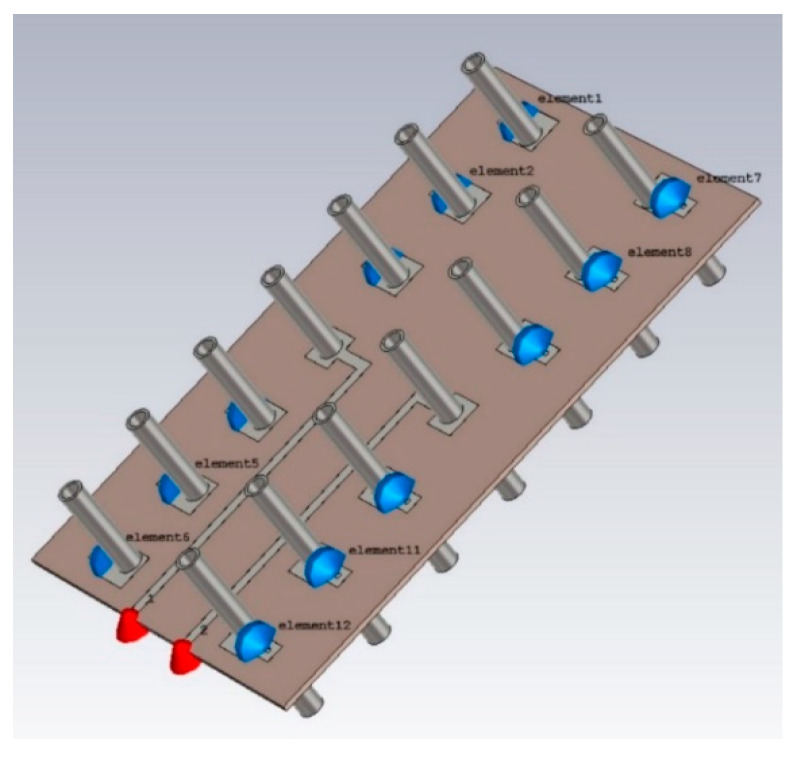
The geometry of the MAMP array in CST; the cylinders are half-wave dipoles connected to passive components and grounded through via holes. The blue dots are the loads (passive components, capacitors, or inductors calculated from SBA), whereas the red dots are the feeding ports to feed the active elements.

**Figure 5 micromachines-15-01351-f005:**
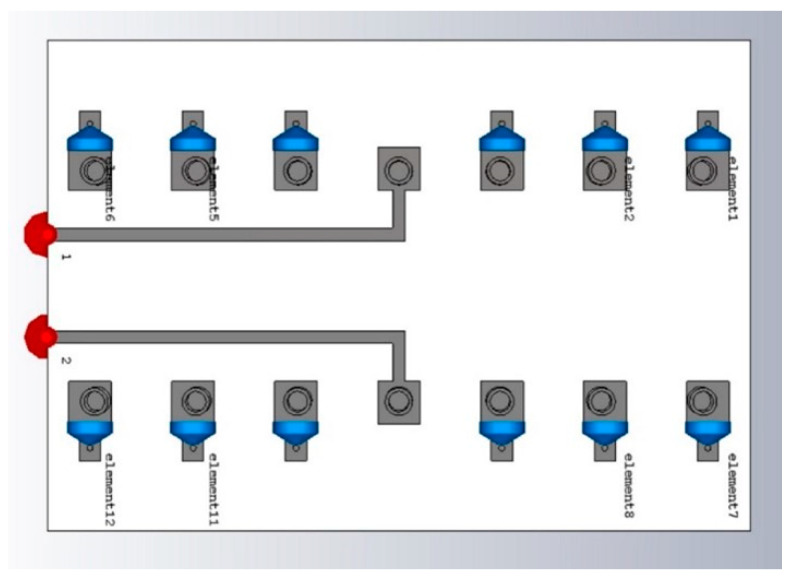
Geometry of the MAMP in CST (**a**) Top view of the structure (**b**) bottom view.

**Figure 6 micromachines-15-01351-f006:**
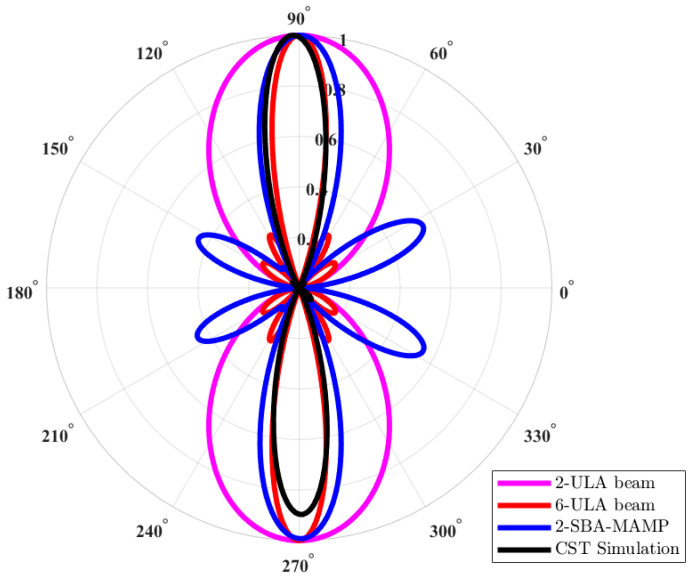
Theoretical and simulated results of radiation patterns of MAMP array; comparison of the radiation patterns where magenta represents a ULA with 2 active elements, red is for the 5 ULA to be compared with the MAMP beam, blue is the calculated beam pattern of MAMP using SBA based on different combination of loads, and black is the radiation pattern of the MAMP array obtained from CST simulations.

**Figure 7 micromachines-15-01351-f007:**
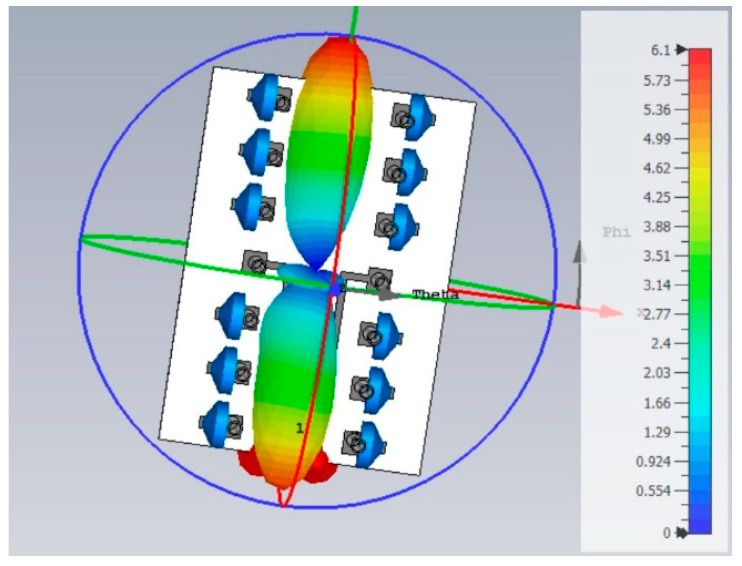
Three-dimensional Radiation pattern of MAMP array obtained from CST simulations.

**Figure 8 micromachines-15-01351-f008:**
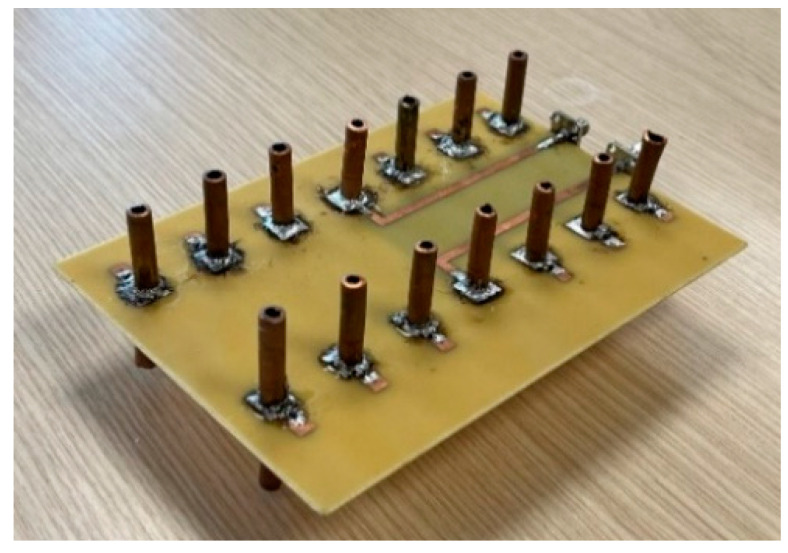
MAMP antenna array prototype.

**Figure 9 micromachines-15-01351-f009:**
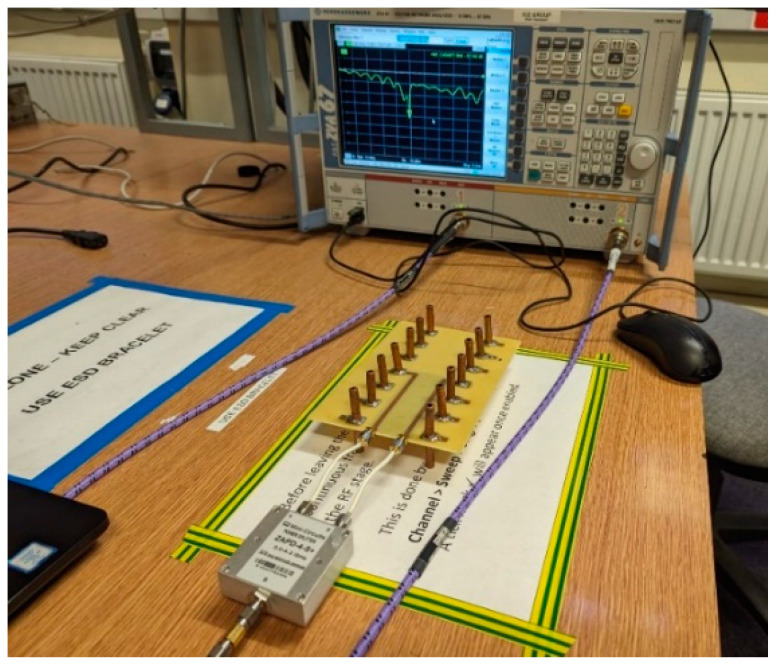
Setup for measuring the S-parameters using a vector network analyzer.

**Figure 10 micromachines-15-01351-f010:**
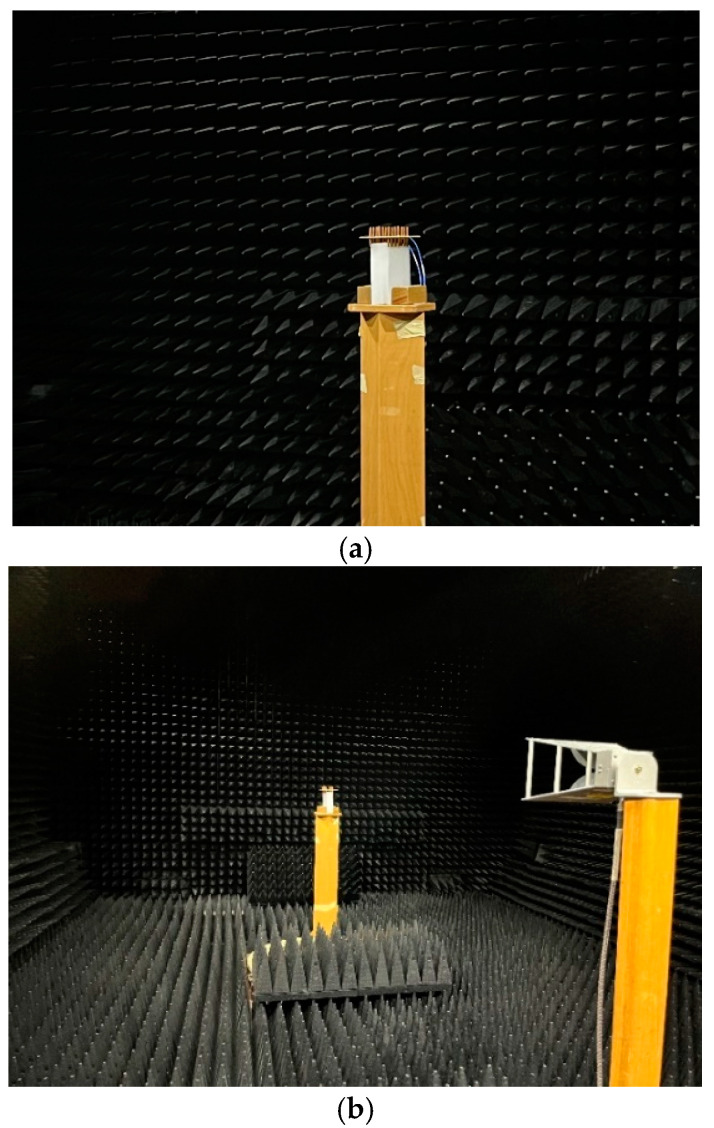
Measuring radiation pattern in the anechoic chamber (**a**) antenna under test (**b**) setup for the measurement of radiation pattern.

**Figure 11 micromachines-15-01351-f011:**
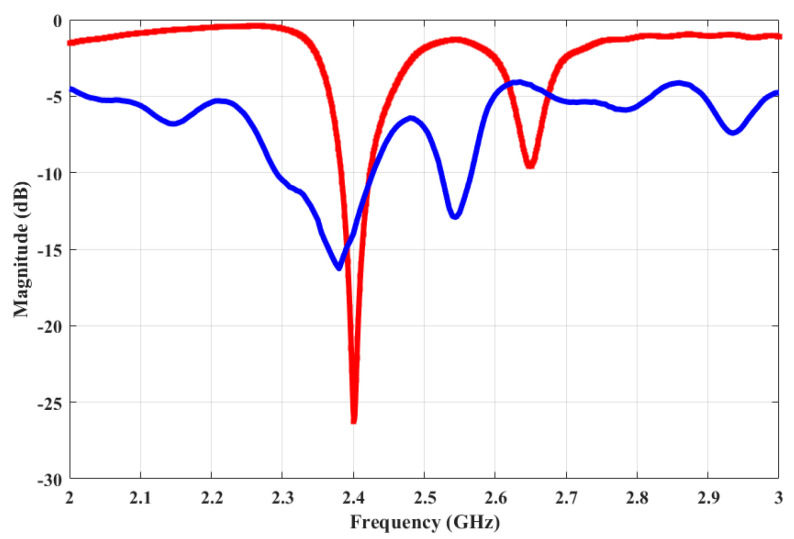
Measured and the simulated S-parameters of the MAMP antenna array. The red curve represents the simulated while blue represents the measured S-parameters of the structure.

**Figure 12 micromachines-15-01351-f012:**
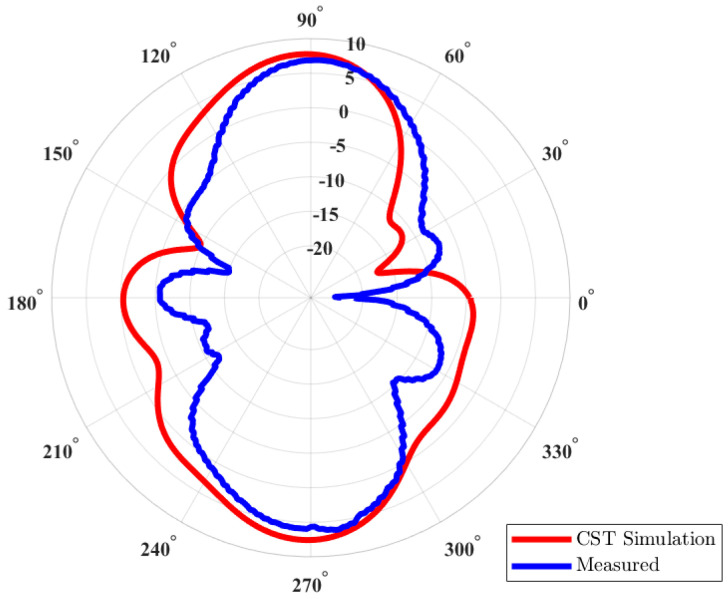
Measured and simulated polar radiation patterns of the MAMP antenna array.

**Figure 13 micromachines-15-01351-f013:**
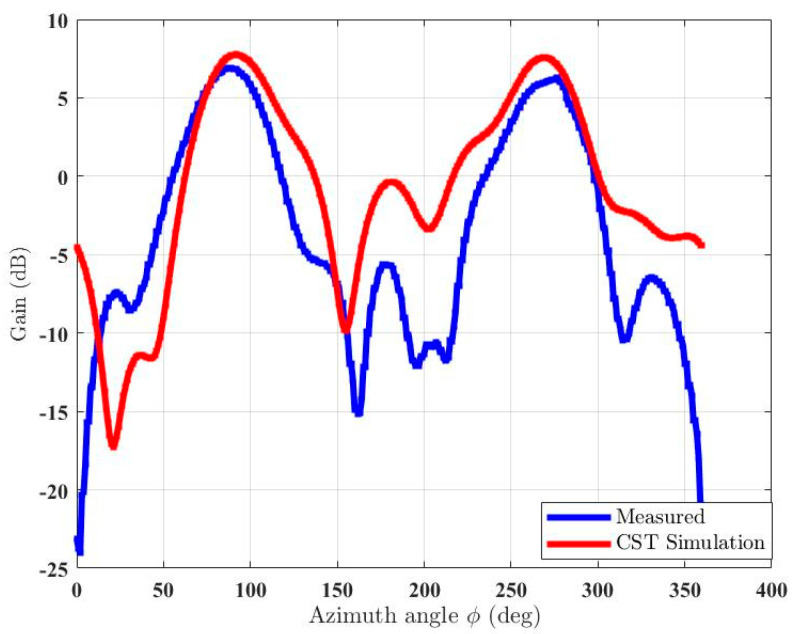
Measured and simulated gain of the MAMP antenna array.

**Table 1 micromachines-15-01351-t001:** Comparison of Proposed MAMP Array Between the Reported Work.

Previous Work	Key Technique	SAMP	MAMP	Radiation Constraints	Load Values	Radiation Efficiency	Prototype Development
Perturbation stochastic methods [4,5,6,7]	Beamforming using ESPARs	Yes	No	No constraints	Complex	No	Only [6]
beam space models [8,9,10,11]	Spatial multiplexing of QPSK symbols	Yes	No	No constraints	Complex	No	Only [11]
Design guidelines and the radiation constraints, quantization, angle of arrival estimation [14,15,16,17,20]	Beamforming using PAAs	Yes	No	Only for single-RF	Complex	No	No
Arbitrary Signal models and quantization [23,24,25,26]	MIMO transmission	Yes	No	Only for single-RF	Complex	No	Only [26]
stochastic beamforming algorithm [18]	Beamforming using PAAs	No	Yes	No constraints	Reactive only	No	No
Design Guidelines [19]	Beamforming using PAAs	No	Yes	For MAMP/partial conditions	Complex	No	No
Stochastic beamforming algorithm [30]	Beamforming using PAAs	No	Yes	No constraints	Reactive only	No	No
[Our work] Stochastic beamforming algorithm	Beamforming using PAAs	No	Yes	Constraints for a whole MAMP array	Reactive only	RE integrated in cost function	Yes

## Data Availability

All data generated or analyzed during this study are included in this manuscript. There are no additional data or datasets beyond what is presented in the manuscript.

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
