# Peer review of "Energy Efficient Multi-Active/Multi-Passive Antenna Arrays for Portable Access Points"

_micromachines, 2024, doi:10.3390/mi15111351_

Round 1

Reviewer 1 Report

Comments and Suggestions for Authors

The author presented a antenna array for portable access points. The paper is written in professional way, backed with in-depth and state-of-the-art literature work. All the concept and methodologies are explained properly with related mathematical expressions, which further strengthen the quality of the manuscript. There is no error found in study and thus it is recommended for publication in its current form.

Author Response

We would like to thank you for your time and efforts to improve the manuscript.

Comments 1: [The author presented a antenna array for portable access points. The paper is written in professional way, backed with in-depth and state-of-the-art literature work. All the concept and methodologies are explained properly with related mathematical expressions, which further strengthen the quality of the manuscript. There is no error found in study and thus it is recommended for publication in its current form.]

Response 1: [We are pleased that you were satisfied with the research work and how it is presented in the submitted manuscript.]

Reviewer 2 Report

Comments and Suggestions for Authors

In this manuscript, the author proposes a novel Multi-Active/Multi-Passive (MAMP) antenna array design, which can be used on portable access points (PAP) to provide wireless connectivity services.The results are interesting. However, some of the comments listed below should be considered before it can be published.

1. In this manuscript, the author proposes an algorithm for calculating the parameters of the MAMP antenna array, but there is no relevant program flowchart in the manuscript. It is recommended to supplement the explanation.

2. For an antenna array, coupling between ports is a very important parameter. Excessive coupling can seriously affect the performance of the antenna. However, I did not see relevant descriptions or solutions to coupling in this manuscript. It is recommended to supplement the explanation.

3. The efficiency of the antenna is also an important parameter for this type of antenna. In this manuscript, only the theoretical radiation efficiency is described, and no actual measured efficiency curves are found. It is recommended to supplement the explanation.

4. In this manuscript, there is a certain discrepancy between the measured and simulated S-parameters of the antenna, and you did not provide relevant explanations. It is recommended to add explanations.

Comments on the Quality of English Language

Minor editing of English language required.

Author Response

1. Summary        
Thank you very much for taking the time to review this manuscript. Please find the detailed responses below and the corresponding revisions/corrections highlighted changes in the re-submitted files. 

Comments 1: [In this manuscript, the author proposes an algorithm for calculating the parameters of the MAMP antenna array, but there is no relevant program flowchart in the manuscript. It is recommended to supplement the explanation.] 

Response 1: Thank you for pointing this out. We agree with this comment. Therefore, we have added the algorithm in the manuscript, please refer to page no. 7, line 265.
“[Algorithm 1 Alternating Optimization Stochastic Beamforming Algorithm
1: function AO-SBA (b, 〖(β_m)〗_(m=1)^M, S, Z, S, τ, Nm, Ter, eps, ω,µ)
2: m ← 0,x_Sc  = 〖[0,...,0]〗^T  ,w ← 〖[1^T,0^T   ]〗^T   
3: while m<M do
4:  m ← m + 1, n ← 0, k ← 0
5:  ã€–err〗_x ← 1/eps, 〖err〗_w ←  1/eps
6:  while n<N_m and 〖err〗_x≥T_er  do
7:     n ← n + 1, x_old ← x_Sc
8:     Create: δ_x ∼ B(1,1/2) with values ±1
9:     x_Sc^+= x_Sc+ β_m δ_x, x_Sc^-= x_Sc- β_m δ_x
10:     L_x^+=L(x_Sc^+  ,w)  ,L_x^-=L(x_Sc^-  ,w)   
11:     ξ_x=1/(2β_m )  (L_x^+- L_x^- )  1⊘δ_x 
12:     x_Sc= x_Sc- τξ_x,〖err〗_x=‖x_Sc-x_old ‖_2/(‖x_old ‖_2+eps)
13: end while
14: while k<N_m and 〖err〗_v≥T_er  do
15:     k ← k + 1, w_old ← w
16:     Create: δ_w ∼ B(1,1/2) with values ±1
17:     w^+=w+β_m δ_w,w^-=w-β_m δ_w    
18:     L_v^+=L(x_Sc  ,w^+ )  ,L_x^-=L(x_Sc  ,w^- )
19:     ξ_w=1/(2β_m )  (L_v^+- L_v^- )1⊘δ_w
20:     w=w-τξ_w,〖err〗_w=‖w-w_old ‖_2/(‖w_old ‖_2+eps) 
21: end while
22: while      real (Z_in^j )>0, do     Given by (3.3) and (3.4)
23: p ← p + 1, jϵ{1,2,3,….,N}
24: Go to function AO-SBA
25:    while 0 < η < 1
26:        p ← p + 1
27:         ω+ µ=1,
28:        η=1- 〖Γ_T〗^2, Γ_T= √(1-  P_rad/P_in )
29:     end while 
30: end while
31: Er(m)  ← L(x_Sc  ,w)            Given by (3.18) and (3.19)
32: end while
33: end function
Ensure: 〖Er〗_(1:M),x_Sc,w,η 
]”
Comments 2: [For an antenna array, coupling between ports is a very important parameter. Excessive coupling can seriously affect the performance of the antenna. However, I did not see relevant descriptions or solutions to coupling in this manuscript. It is recommended to supplement the explanation.]
Response 2: The comment is considered and the related correction in indicated at page 4, line 134. 
‘‘[Z_L is a matrix representing the source resistance at all the active elements and the load values of parasitic elements. Z_MM  is the mutual coupling matrix and represents the coupling among the elements of the whole array. It shows the self-impedance and the mutual impedance of the neighboring elements and can be seen in equation (2). v_M is the input voltage at the feeding active elements of the arrays and at the parasitic elements of the array, which is zero. Equation (1) can be expanded for the MAMPs as given in [25]:

[â– (Z_11+x_1&Z_12&⋯@Z_21&Z_22+R_1&⋯@Z_31&Z_32&⋯@â‹®&â‹®&⋱@â‹®&â‹®&⋱@â‹®&â‹®&⋯@Z_M1&Z_M2&⋯)â– (Z_1M@Z_2M@Z_3M@â‹®@â‹®@â‹®@Z_MM+x_M )][â– (i_1@i_2@i_3@â‹®@â‹®@â‹®@i_M )]=[â– (0@v_1@0@â‹®@0@v_N@0)]                     (2)

From (2), the variable x_M is the m-th load value of the respective parasitic element of the array, where i_m  is the current at the m-th element and v_j is the voltage at the j-th active element in the MAMP array, and j ϵ {1,2,3,….,N}. Since each set of single-active / multi-passive (SAMP) is placed at half wavelength distance from every other SAMP, which means that the active elements are half wavelength from each other thus reducing unwanted coupling effects. Whereas the parasitic elements are placed in the close vicinity to get the more coupling. The performance of the multiport antenna array can be analyzed by using the input impedance of the ports. The input impedance connected to the RF chains incurring the effect of mutual coupling from neighboring elements can be obtained from (2):

Z_in^j=∑_(m=1)^Mâ–’Z_jm   i_m/i_j ,jϵ{1,2,3,….,N}                                                     (3)
where Z_in^jrepresents the input impedance at the j-th port,〖 Z〗_jm  is the mutual coupling between the j-th active element and the m-th element in the MAMP array and i_j represents the current at the j-th active element in the MAMP array. ]’’

Comments 3: [The efficiency of the antenna is also an important parameter for this type of antenna. In this manuscript, only the theoretical radiation efficiency is described, and no actual measured efficiency curves are found. It is recommended to supplement the explanation.]
Response 3: [The radiation efficiency of an antenna can be measured using some specific reverberation chambers only, which we don’t have the access at the moment. Although the measured efficiency can be calculated using the total reflection from the feeding ports to see the array performance.]
Comments 4: [In this manuscript, there is a certain discrepancy between the measured and simulated S-parameters of the antenna, and you did not provide relevant explanations. It is recommended to add explanations.]
Response 4: [Explanation to this comment is added at Page 16, line 481. 
Simulated and measured S-parameters show a good match as shown in Fig. 10. There is a slight shift in the resonant frequency of prototyped antenna as shown in the measured S-parameters, but it is still well matched for the desired frequency i.e., 2.4GHz. A slight discrepancy is due to use of wideband power splitter while doing measurements.]

Reviewer 3 Report

Comments and Suggestions for Authors

The authors presented "Energy Efficient Multi-Active / Multi-Passive Antenna Arrays for Portable Access Points" to achieve better network connectivity. A detailed analysis has been conducted to validate the system's performance. However, the reviewer has some minor suggestions to further improve the manuscript.

1.       In the Introduction, the term "PAP" (Portable Access Points) is introduced without a clear initial definition. Although it's explained later, adding a brief description right after first mentioning the acronym would aid reader comprehension.

2.       The equations and mathematical expressions are well-presented, though some would benefit from a brief introductory sentence to explain their purpose. For example, introducing Equation (1) with a sentence like, "The generated currents can be derived using the following equation," could help readers follow the mathematical progression.

3.       In Figure 8, while the prototype is labeled well, a brief note about the fabrication process and the choice of materials (e.g., copper dipoles on FR substrate) could add clarity, especially for readers unfamiliar with these specifics.

4.       The paper maintains a formal tone, but a few grammatical tweaks could enhance readability. For example, phrases like "solve the problem by mounting small cell base stations on it" could be more fluid as "address this issue by deploying small cell base stations."

5.       The conclusion effectively summarizes the paper but could expand on future work or possible applications. Discussing how this technology might be integrated into commercial UAV systems, for example, could provide a broader context.

Author Response

1. Summary

Thank you very much for your efforts and suggestions. We have considered all the comments and addressed to them accordingly. Please find the detailed explanation to all the comments below.

Comments 1: [In the Introduction, the term "PAP" (Portable Access Points) is introduced without a clear initial definition. Although it's explained later, adding a brief description right after first mentioning the acronym would aid reader comprehension.]

Response 1: [The comment is addressed as suggested. The PAPs are explained in the beginning and the 1st paragraph of the introduction. Page 1, line 39]

Comments 2: [The equations and mathematical expressions are well-presented, though some would benefit from a brief introductory sentence to explain their purpose. For example, introducing Equation (1) with a sentence like, "The generated currents can be derived using the following equation," could help readers follow the mathematical progression.]

Response 2: [Thank you for the suggested correction. All the equations in the revised manuscript are explained as suggested.]

Comments 3: [In Figure 8, while the prototype is labeled well, a brief note about the fabrication process and the choice of materials (e.g., copper dipoles on FR substrate) could add clarity, especially for readers unfamiliar with these specifics.]

Response 3: [Page 13, line 442. The explanation to figure 8 is added in the paragraph.

‘‘The MAMP antenna array prototype is developed at University College London (UCL). Fig. 8 shows the developed antenna, where copper dipoles are embedded on a PCB where, substrate is FR-4. The copper monopoles are soldered on each side of the PCB board, the parasitic elements are soldered and grounded through SMD passive components and using via holes. Active dipoles are fed through microstrip lines and SMA connectors. The developed prototype is tested using a network analyzer to see the resonance of the antenna. The Fig. 9 shows the setup for the measurements of the S-parameters.’’]

Comments 4: [The paper maintains a formal tone, but a few grammatical tweaks could enhance readability. For example, phrases like "solve the problem by mounting small cell base stations on it" could be more fluid as "address this issue by deploying small cell base stations."]

Response 4: [The correction is made as suggested.]

Comments 5: [The conclusion effectively summarizes the paper but could expand on future work or possible applications. Discussing how this technology might be integrated into commercial UAV systems, for example, could provide a broader context.]

Response 5: [The future work and possible extension of this work is added at Page 17, line 517.

‘‘The development of MAMP antenna array for MIMO systems could be a possible extension of this work with more focus on finding the currents at each element of MAMP rather than finding the loads at parasitic elements. Current work considered ideal point sources for the calculation of coupling among the elements, the new focus can be the distribution of currents at each element and fining the efficiency of the MAMP antenna array system.’’]

Round 2

Reviewer 2 Report

Comments and Suggestions for Authors

Accept in present form